# Pareto rules for malaria super-spreaders and super-spreading

Laura Cooper[1,2], Su Yun Kang[3], Donal Bisanzio[3,4,5], Kilama Maxwell[6], Isabel Rodriguez-Barraquer [7,8], Bryan Greenhouse[8], Chris Drakeley [9], Emmanuel Arinaitwe[6,9], Sarah G. Staedke[9], Peter W. Gething[3], Philip Eckhoff[10], Robert C. Reiner Jr. [11,12], Simon I. Hay [11,12], Grant Dorsey[8], Moses R. Kamya[13], Steven W. Lindsay[14], Bryan T. Grenfell[1,15,16] & David L. Smith [11,12]

Heterogeneity in transmission is a challenge for infectious disease dynamics and control. An 80-20 "Pareto" rule has been proposed to describe this heterogeneity whereby 80% of transmission is accounted for by 20% of individuals, herein called super-spreaders. It is unclear, however, whether super-spreading can be attributed to certain individuals or whether it is an unpredictable and unavoidable feature of epidemics. Here, we investigate heterogeneous malaria transmission at three sites in Uganda and find that super-spreading is negatively correlated with overall malaria transmission intensity. Mosquito biting among humans is 90-10 at the lowest transmission intensities declining to less than 70-30 at the highest intensities. For super-spreaders, biting ranges from 70-30 down to 60-40. The difference, approximately half the total variance, is due to environmental stochasticity. Super-spreading is thus partly due to super-spreaders, but modest gains are expected from targeting super-spreaders.

[1] Department of Ecology and Evolutionary Biology, Princeton University, Princeton, NJ, USA. [2] Department of Veterinary Medicine, Cambridge University, Cambridge, UK. [3] Oxford Big Data Institute, Li Ka Shing Centre for Health Information and Discovery, University of Oxford, Oxford, UK. [4] RTI International, Washington, DC, USA. [5] Epidemiology and Public Health Division, School of Medicine, University of Nottingham, Nottingham, UK. [6] Infectious Diseases Research Collaboration, Kampala, Uganda. [7] Department of Epidemiology, Johns Hopkins University, Baltimore, MD, USA. [8] Department of Medicine, University of California, San Francisco, CA, USA. [9] London School of Hygiene & Tropical Medicine, London, UK. [10] Institute for Disease Modeling, Bellevue, WA, USA. [11] Department of Health Metrics Sciences, School of Medicine, University of Washington, Seattle, WA, USA. [12] Institute for Health Metrics and Evaluation, University of Washington, Seattle, WA, USA. [13] School of Medicine, Makerere University College of Health Sciences, Kampala, Uganda. [14] School of Biological and Biomedical Sciences, Durham University, Durham, UK. [15] Woodrow Wilson School of Public and International Affairs, Princeton University, Princeton, NJ, USA. [16] Fogarty International Center, National Institutes of Health, Bethesda, MD, USA. Correspondence and requests for materials should be addressed to D.L.S. (email: smitdave@uw.edu)

Heterogeneity shapes infectious disease epidemiology and transmission. Understanding the causes and consequences of heterogeneity is important for analysis of infectious disease data and for determining target intervention coverage levels for control. A "Pareto rule" has been proposed for many infectious disease systems: 80% of infectious disease transmission is concentrated on 20% of hosts[1]. Such heterogeneity, often called "super-spreading," has drawn interest because the efficiency of disease control could be dramatically improved if it were possible to identify and target individuals who account for most of transmission, who are sometimes called "super-spreaders"[1–6], a topic on which mathematics and mechanistic models have provided some of the most important insights[7–9]. Here, we test the proposed Pareto rule and provide quantitative entomological estimates of malaria super-spreaders and super-spreading using mosquito count data from a large study of malaria in Uganda.

Epidemiological theory has identified several quantitative effects of biting heterogeneity on malaria epidemiology and transmission. Heterogeneity in biting rates among individuals could amplify the transmission of pathogens as they invade a population, facilitating establishment of stable endemic transmission[3]. Heterogeneity could affect the functional relationships among the metrics used to measure malaria transmission intensity[10–12]. Heterogeneous exposure could be an important confounder in epidemiological studies, as individuals with higher rates of infection and clinical disease may also develop immunity more rapidly[13]. Since malaria immunity is only partially protective, a correlation between exposure and immunity could produce a spurious correlation between immunity and disease[14–16]. Finally, if it were possible to target those who are bitten most, heterogeneous biting could allow for more efficient malaria control and elimination[1]. These notions are motivated by mathematical theory, supported by the observation that mosquito counts tend to follow a negative binomial distribution, and are consistent with both a mechanistic understanding of mosquito behavior and studies describing heterogeneous transmission.

Hypotheses about heterogeneity and transmission—scaling relationships, confounding, and targeting—relate to the fraction of heterogeneous biting that is consistently and predictably associated with households or individuals, as opposed to other sources of variability: primarily seasonality and environmental heterogeneity. Mosquito biting patterns, human exposure to malaria parasites, and malaria infection risk vary seasonally and are demonstrably heterogeneous when examined using molecular[17–20] or statistical techniques[21], and have been associated with several factors[3]. Heterogeneity in biting over time can also arise from factors such as wind and human activity patterns, hereafter referred to as environmental stochasticity. Notably, heterogeneity in transmission can also be affected by differences in the infectiousness of individuals[22], which may be unrelated to biting rates.

Heterogeneity in transmission can be quantified and discussed in terms of super-spreaders and super-spreading. Super-spreading for malaria describes heterogeneous transmission in which some hosts would infect more hosts in the next generation than expected by chance alone. Super-spreading can happen if some hosts are more infectious than others; for malaria, this would mean having gametocyte densities that were higher, or that were carried longer, that were accompanied by gametocyte-stage blocking immunity that was lower than the population averages. Alternatively, malaria super-spreading can happen because a host is bitten by more vectors. A common finding is that mosquito counts follow negative binomial distributions, which suggests super-spreading through heterogeneous biting is common for malaria.

Super-spreaders are hosts who would consistently be found super-spreading; they are either consistently more infectious (e.g., with high gametocytemia or low immunity), or they are consistently bitten more by mosquitoes. Since it takes two bites to transmit malaria parasites from a human back to other humans, populations with super-spreaders have built-in correlations that amplify transmission; super-spreaders are both more likely to become infected and more likely to infect others[23–27]. To identify and target super-spreaders, it is necessary to identify those individuals who are consistently bitten more than others.

Here, we examine the relationship between super-spreaders and super-spreading through a study of heterogeneous biting using mosquito count data collected during entomological surveillance in a large malaria study in Uganda, where entomological was collected on focal individuals in households. Using these data, we computed the Pareto fractions—the proportion $X$ of the population that accounts for a proportion $1-X$ of all counts. We also estimated the proportion of variance explained by biting weights, seasonality, and environmental stochasticity. We show that the 80-20 rule for super-spreading holds overall, but we also show that the rule varies with transmission intensity. At the lowest transmission intensities in any month, 10% of individuals get 90% of all bites, but at the highest transmission intensities, 30% of individuals get 70% of all bites. By identifying super-spreaders and re-examining the Pareto rules, we show that transmission is less concentrated: at the lowest transmission intensities that 30% of individuals get 70% of all bites, and at the highest transmission intensities, 40% of individuals get 60% of all bites. Super-spreading through heterogeneous biting on super-spreaders is thus important for malaria transmission dynamics and control, but environmental stochasticity also plays a role accounting for about half of super-spreading.

## Results

**Seasonality and transmission**. The study reports on data from Walukuba, Jinja District and Kihihi, Kanungu District over 42 months, and Nagongera, Tororo District for 69 months. The study in Tororo captured a sharp decline in mosquito densities following the implementation of an indoor residual spraying program in Tororo District by the Uganda National Malaria Control Program[28], with spraying in December 2014, corresponding to the beginning of year three in Fig. 1. Spraying at regular intervals continued through the end of the study. The average annual entomological inoculation rate (EIR) at these three sites was lowest in Walukuba, Jinja District (annual EIR ≈ one infectious bite per person per year, *ibppy*), intermediate in Kihihi, Kanungu District (annual EIR ≈ nine *ibppy*), and highest in Nagongera, Tororo District (annual EIR ≈ 85 *ibppy*) before the IRS spraying program. These estimates of the EIR are slightly lower than found for 2011[29]. The present analysis also quantified the temporal trend in average exposure with an irregular seasonal pattern across years as well as the sharp decline in mosquito counts after the IRS spraying program. The expected number of anopheline mosquitoes caught was denoted $S_{d,n}$ for day $d$ at the $n$th site (see Methods section, Fig. 1).

**Super-spreaders and super-spreading**. The mosquito counts in our study are described well by negative binomial distributions; the variance of the counts data was consistently much higher than the mean[30], consistent with super-spreading. To quantify super-spreaders vs super-spreading, we developed a statistical model for these negative binomial distributions (see Methods), and we have summarized the patterns using Pareto fractions (see Methods). The Pareto fraction for super-spreading is defined as the proportion $X$ of the total HBR or EIR received by the proportion $1$

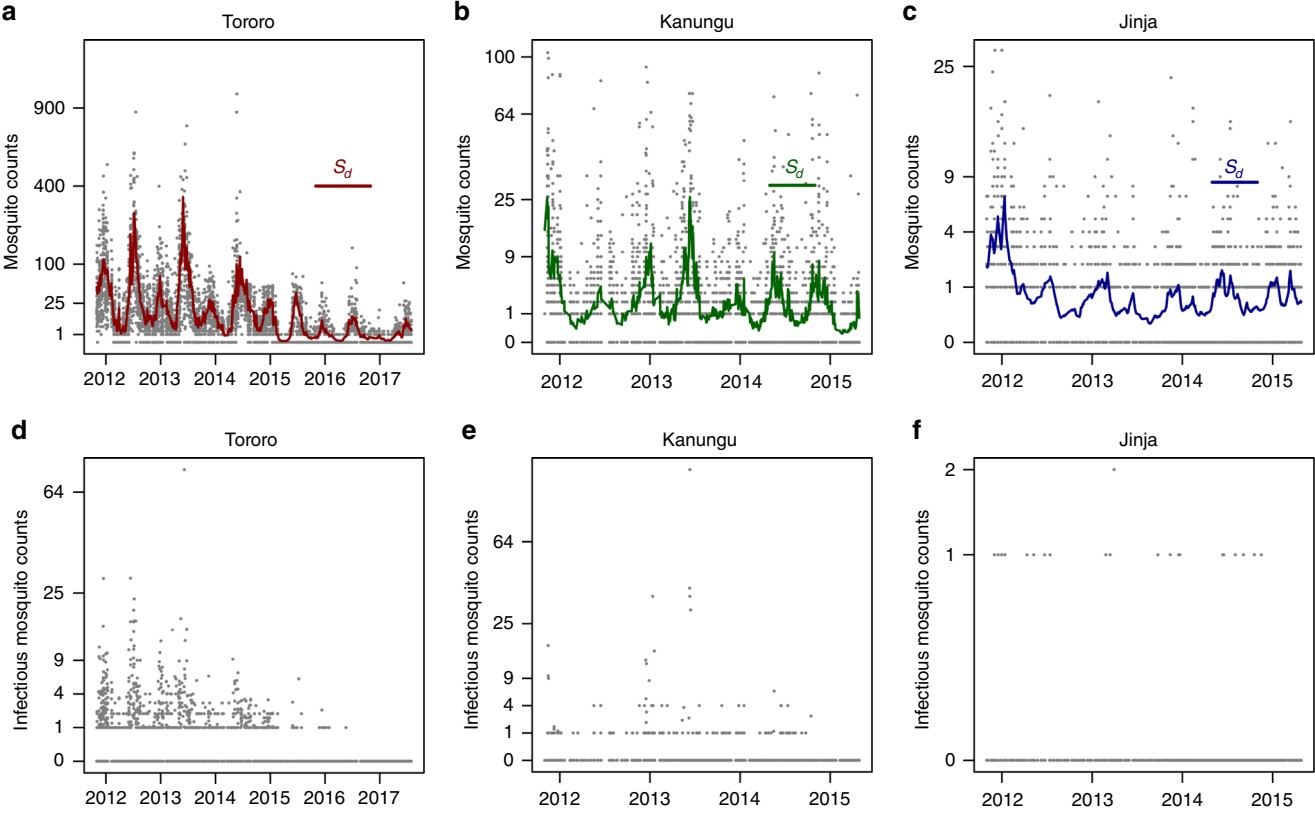

**Fig. 1** Mosquito counts and the modeled daily expectation. In all plots, the y-axis shows the square root of the observations. The total anopheline mosquito counts and daily expectation $S_d$ for **a** Tororo, **b** Kanungu, and **c** Jinja. The number of sporozoite-positive mosquitoes for **d** Tororo, **e** Kanungu, and **f** Jinja

$-X$ of households (i.e., the point where the empirical cumulative distribution function intersects the line $1-X$). The analysis to compute Pareto fractions was modified for super-spreaders. The same algorithm is repeated but only after re-ordering the observations by the biting weights ($\omega_{h,n}$, see Methods). In the modified Pareto analysis, is possible those who tend to get the most bites (on average) will sometimes account for <50% of all bites.

**Household biting weights and environmental stochasticity**. Our data allow us to examine super-spreaders and super-spreading associated with heterogeneous biting by mosquitoes. To assess how much super-spreaders contributed to super-spreading, the statistical analysis estimated household biting weights for each household at each site, denoted $\omega_{h,n}$, which are interpreted as the ratio of a household's expectation to the overall expectation for the population (Fig. 2a). Households with higher weights tend to get more bites by mosquitoes and thus contribute more to transmission; the weights are the basis for identifying super-spreaders. The remaining variance is explained by environmental stochasticity and measurement errors ($\varepsilon$), which is Gamma distributed (see Methods). In the statistical model, counts on households are thus a compound Gamma-Poisson mixture distribution $X_{d,h,n} \sim \text{Poisson}(S_{d,n}\, \omega_{h,n}\, \varepsilon)$. We have used this analysis both to quantify super-spreading and to weigh the importance of super-spreaders vs. super-spreading.

The challenge for understanding the consequences of heterogeneity on transmission and for targeting and other epidemiological phenomena is that the observed distribution of catch counts could occur with or without super-spreaders; the potential contribution of super-spreaders can be represented along a spectrum. One hypothetical process, conceptually located at one extreme of this spectrum, is that after accounting for temporal

heterogeneity and sampling noise, biting weights would account for all the remaining variance in HBR; super-spreading would be entirely attributable to super-spreaders (i.e., it is associated with $\omega$). At the other extreme of this spectrum, daily environmental stochasticity would account for all remaining variance, leaving nothing explained by super-spreaders. To put it another way, the negative binomial distributions could arise either because the biting weights are Gamma distributed, or because the measurement errors are Gamma distributed, or both. Points along the spectrum represent different fractions of heterogeneity apportioned to biting weights (i.e., super-spreaders) versus environmental stochasticity (super-spreading without super-spreaders).

Our data and analysis allow us to evaluate where these populations are located on this continuum by quantifying what fraction of the variance in biting is explained by our estimated daily seasonal expectation for the HBR, and the HBR with the biting weights (i.e., $S_{d,n}$ vs. $S_{d,n}\omega_{h,n}$, see Methods). We also simulated counts by drawing from a Poisson distribution with mean $S_{d,n}\omega_{h,n}$ (i.e., but not $\varepsilon$) to get an estimate of the sampling variance. In this way, we estimated the proportion of variance associated cumulatively with seasonality, biting weights, sampling variance, and environmental stochasticity and measurement errors (Fig. 2b).

Approximately half of the variance was attributable to the combined effects of seasonality and biting weights. After accounting for seasonality, the proportion of the variance explained by biting weights ranked in reverse order to biting intensity; biting weights accounted for the highest fraction of the variance at the lowest-intensity site.

**Pareto rules for super-spreading**. Using catch data for the daily human biting rate (HBR), we found that overall, these sites were

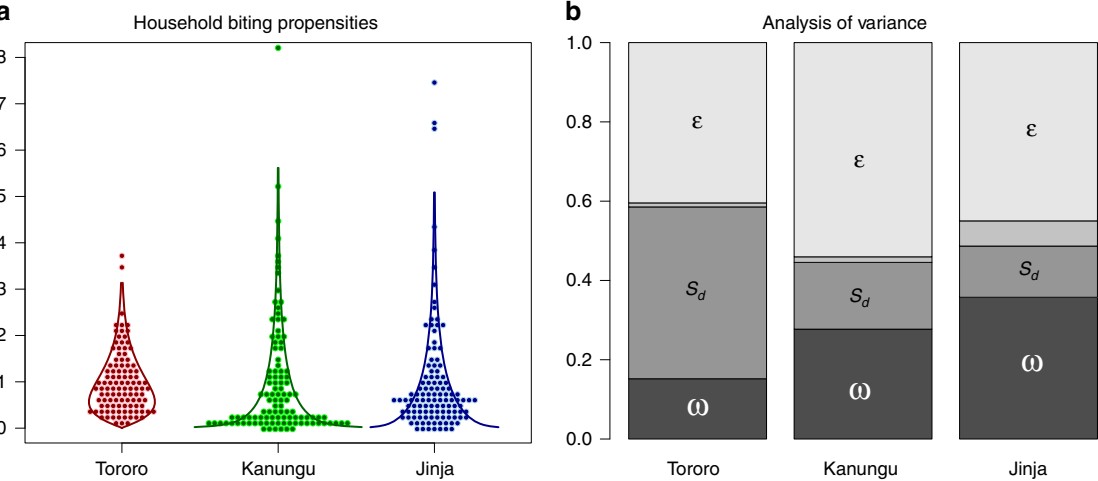

**Fig. 2** Biting weights and results of the variance components analysis. **a** Distribution of biting weights (i.e., the points are the fitted values $\omega_{h,n}$ from the fitting procedure) and a Gamma distribution fitted to describe the points using MLE (solid lines, plotted to the 99th quantile) for Jinja, Gamma(1.03, 1.03); Kanungu, Gamma(0.71, 0.71); Tororo, Gamma(2.41, 2.41). **b** Proportion of variance explained by biting weights ($\omega$), seasonality in the HBR ($S$), the estimated sampling variance (the smallest sliver), and environmental stochasticity and measurement errors ($\varepsilon$)

close to 80-20, but it was not true that 20% of the population would receive 80% of all bites across settings and seasons. Overall, the Pareto fractions (i.e., the fraction $1-X$ of the counts that were concentrated on the fraction $X$ of the population) were 82:18 for Tororo, 84:16 for Kanungu, and 85:15 for Jinja (Fig. 3). By way of contrast, the Pareto fraction for large draws from the Poisson distribution are approximately 68:32. For EIR, which measures exposure to parasites, the Pareto fractions were higher: 89:11 for Tororo, 97:3 for Kanungu, and 99:1 for Jinja (Fig. 3). The utility of the Pareto analysis is at least partly limited when the data are dominated by zero counts, as they are for HBR at the lowest intensities and for EIR at two of the sites (Fig. 3). When there are only a handful of positive observations, the Pareto analysis reports that fact almost tautologically.

For both HBR and EIR, there was a strong negative correlation between the logged monthly mean counts and the Pareto fraction for data collected during that month (Fig. 4b, e). For HBR, the trendline for Pareto fractions month by month varied from above 90-10 for daily HBR rates of <0.3 to approximately 70-30 for daily HBR of 175 (Fig. 4b). The Pareto fractions for EIR varied from well above 90-10 down to 80-20 (Fig. 4e). These data suggest that the proposed 80-20 rule does not hold across the spectrum of transmission. Instead, biting becomes less heterogeneous overall as biting intensity increases.

**Pareto analysis for super-spreaders**. The Pareto analysis for super-spreaders also suggests biting weights accounted for only a fraction of the total heterogeneity. Environmental stochasticity reshuffled the identities of the households with the most mosquitoes each month, lowering the estimated degree of aggregation. Overall, the Pareto fractions for the HBR were substantially lower: 59:41 for Tororo, 73:27 for Kanungu, and 72:28 for Jinja (Fig. 3). For the monthly HBR data and for the monthly EIR data, the Pareto fractions ranged from 70:30 down to 60:40 (Fig. 4b, e) and were negatively correlated with transmission intensity.

When examining super-spreaders, it may not be true that the top 50% account for at least 50% of the bites in every month. Here, in 1 month, the Pareto fractions for bites were less than 50-50 for 3 months in Tororo. For sporozoite-positive bites, there were 4, 1, and 3 months where Pareto rules were less than 50-50 for Jinja, Kanungu, and Tororo, respectively (the outliers in Fig. 3, which are more apparent in Fig. 4b, e).

The analysis for super-spreaders shows how much an analysis for super-spreading overestimates the proportion of exposure that is targetable. Metrics for super-spreading are not wholly targetable because they summarize transmission for a different set of highly exposed households each month. Because of environmental variability, the Pareto statistics for super-spreaders necessarily indicate less aggregation than for super-spreading. The statistics for super-spreaders are a better measure of the likely impact of targeting.

**Super-spreaders and transmission**. Theory suggests that heterogeneous biting through super-spreaders could play an important role in sustaining transmission; populations are more readily invaded by malaria parasites in models where biting is concentrated on a few individuals. Heterogeneous biting amplifies transmission between mosquitoes because those humans who are most frequently bitten are most likely to become infected, which leads to a greater expectation of infections in other mosquitoes[23–27]. Putting this relationship into context for understanding transmission dynamics and control, transmission is amplified by a factor that is related to the coefficient of variation of the distribution of these biting weights, $\alpha$. When biting is heterogeneous and malaria is rare, correlations in mosquito feeding on humans who are most likely to have been infected amplify transmission so that the criterion for invasion would be $R_0 > 1/(1+\alpha)$[3]. The effect size of heterogeneity on invasion is thus equal to $1+\alpha$, where $\alpha$ is the squared coefficient of variation of biting rates.

For heterogeneity to amplify transmission, differences in biting rates must persist long enough for correlations between consecutive infectious bites to be realized, so it is not the overall shape of the biting distribution that matters, but average differences among individuals in their biting rates over the infectious period. For super-spreaders and super-spreading, the relevant quantity is the average ratio for each household of the observed count to the expected site-wide value for that day in a population, what we have called biting weights ($\omega$). In our notation, $\alpha$ is simply the variance of the distribution of biting weights, $V_n$, so transmission would be amplified by a factor $1 + V_n$[3]. We fit Gamma distributions to the estimated biting weights at each site using maximum likelihood estimation, $\omega_{h,n} \sim$ Gamma $(\theta_n, \theta_n)$ which were thus constrained to have a mean of one and

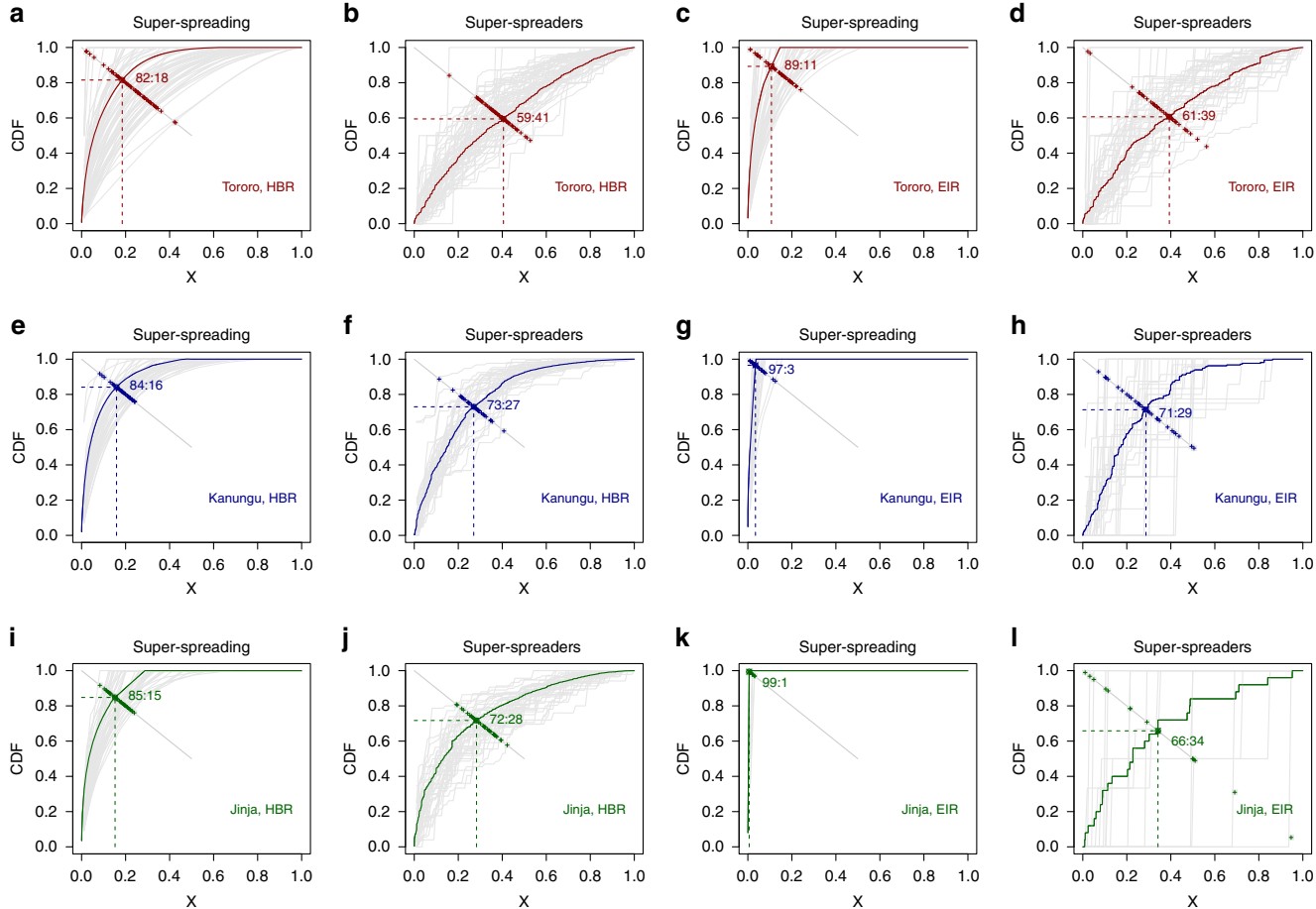

**Fig. 3** The Pareto analysis and fractions by site for super-spreaders or super-spreading for all (HBR) or sporozoite-positive (EIR) anopheline mosquitoes. The data are sorted naturally (for super-spreading) or by the biting weight (for super-spreaders). The point where the CDF crosses the line 1—X is the Pareto fraction. The gray lines in the background show the analysis for each month. The colored line shows the CDF for all the data for each site. For emphasis, the Pareto fraction for each month was also plotted in color. Note that for some months, super-spreaders have a Pareto fraction that is less than 50-50; the houses that tend to get the most bites need not account for half the bites in any particular month. The analyses are shown for Tororo (**a**–**d**); Kanungu (**e**–**h**); and Jinja (**i**–**l**) for super-spreading (**a**, **c**, **e**, **g**, **i**, **k**) and for super-spreaders (**b**, **d**, **f**, **h**, **j**, **l**); for all anophelines (**a**, **b**, **e**, **f**, **i**, **j**), and for sporozoite-positive anophelines (**c**, **d**, **g**, **h**, **k**, **l**)

both variance and coefficient of variation $V_n \approx 1/\theta_n$ (Fig. 3a). Using the variance of the biting weights, heterogeneous biting alone would amplify transmission during an invasion by a factor of 2.6 in Jinja, by 2.73 in Kanungu, and by 1.46 in Tororo. Notably, theory also suggests the basic reproductive number increases with human population density and declines with heterogeneity[3], so the general rule for amplification may not apply to Tororo, which has rural populations with the highest transmission intensities.

## Discussion
Though many studies have examined the properties of mosquito counts within a site over time[31], this study describes the distribution of relative entomological risks for malaria transmission among households at multiple sites across a range of mosquito densities and transmission levels and through multiple seasons. We have shown that some of the observed heterogeneity is predictably associated with individual households over time. In this study, it was not generally true that 20% of the houses had 80% of the mosquitoes. While the variance of these distributions increases with the mean, biting becomes more evenly distributed as transmission intensity increases. The Pareto fraction for mosquito counts varied from 98-2 (i.e., 2% of houses had 98% of mosquitoes) at low intensity to only 57-43 at high intensity. The

trend was clearly negative with respect to transmission intensity, ranging from 90-10 to 70-30. The Pareto fractions for infectious mosquito counts were largely more concentrated than 80-20, but the fractions were also negatively correlated with transmission intensity. One ecological explanation for this negative trend may be that when mosquito numbers are low there are few breeding sites overall and even fewer sites close to settlements. An important question is how well these relationships hold across vector species and the broad range of ecological settings that characterize transmission of malaria and other vector-borne diseases[32,33]. It should be noted, however, that the study directly tested the infectivity of up to 50 mosquitoes from each household per night, but that we relied on extrapolation to quantify higher mosquito numbers in those samples. As a result of that modeling, our analysis for Tororo and for a few months in Kanungu may not have accurately captured outlier occurrence in months with the highest density of infectious mosquitoes.

Our study found strong support for super-spreading, but weaker support for the role of super-spreader households in malaria transmission. Some of the variability in biting was associated with particular households over the study, but our analysis shows that environmental stochasticity accounts for approximately half of the variance. Despite marked differences among households in their propensity to have mosquitoes, the

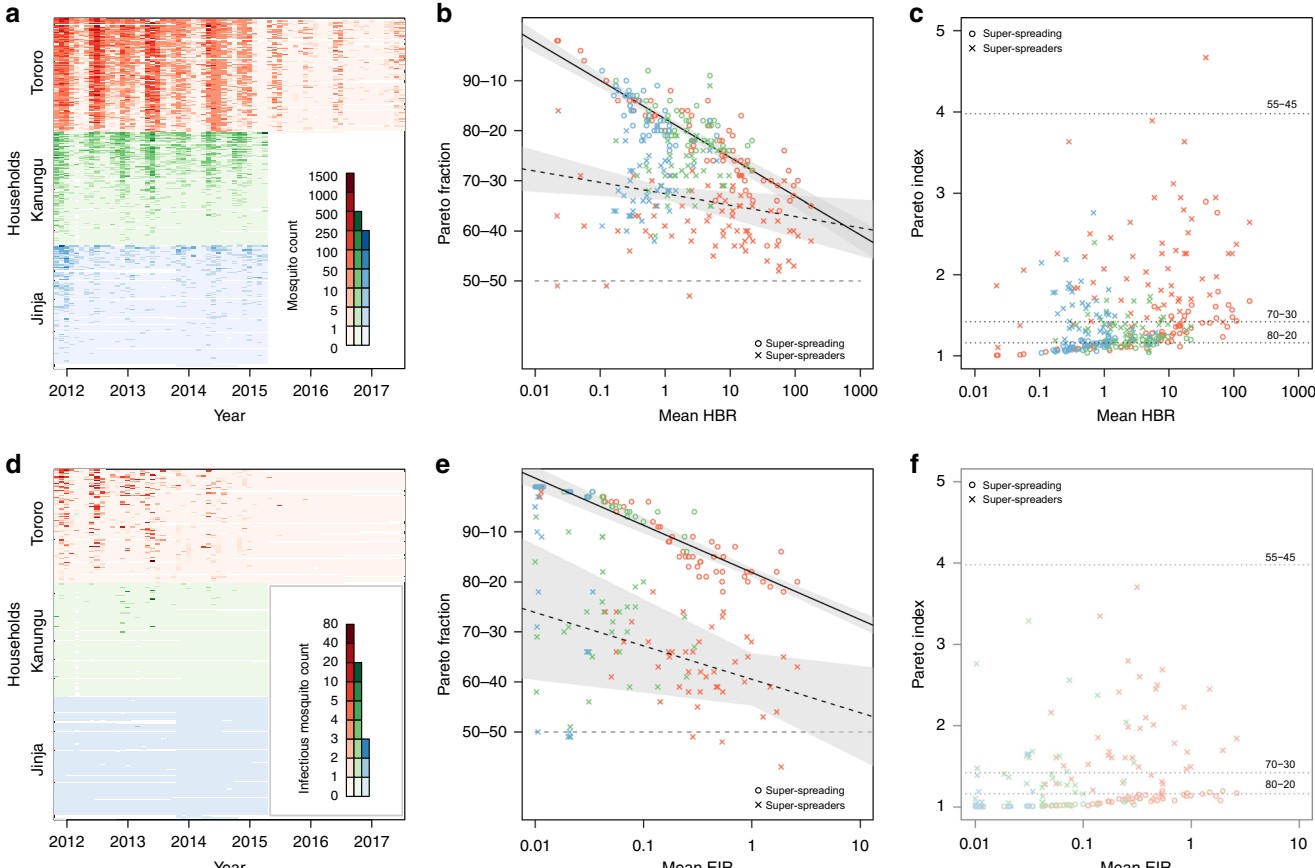

**Fig. 4** The counts by household and summary of the Pareto analysis for anopheline mosquito counts. Top row shows analysis for all anophelines, and the bottom for sporozoite-positive anophelines. **a** Anopheline mosquito catch counts by month and household (each household is on one line), sorted within each site by the median counts for each household. The study ended after 42 months for Jinja and Kanungu; 69 months of data from Tororo are presented here. Darker colors indicate higher counts. **b** Monthly Pareto fractions (e.g., 0.9 is 90-10) for super-spreading (circles) and super-spreader (x's) by mean monthly HBR. Linear fit lines for super-spreading (solid) and super-spreaders (dashed) are shown with confidence intervals in gray. **c** The Pareto index plotted vs. the logged mean monthly HBR for both super-spreading (circles) and super-spreaders (x's). The dashed lines are values of the Pareto index that give the 80-20, 70-30, and 55-45 distributions. The range is restricted to 1–5 (outliers are identifiable in panel **b**). **d** Sporozoite-positive anopheline mosquito catch counts by month and household (each household is on one line), sorted within each site by the median counts for each household. Darker colors indicate higher counts. **e** Monthly Pareto fractions for super-spreading (circles) and super-spreaders (x's) by mean monthly EIR. Linear fit line for super-spreading (line) and super-spreaders (dashed) are shown with confidence intervals in gray. **f** The Pareto index plotted vs the logged mean monthly EIR for both super-spreading (circles) and super-spreaders (x's). As in (**c**), the range is restricted to 1–5 (outliers are identifiable in panel **e**)

identities of the houses receiving the most bites changed over time due to other factors, broadly lumped together under the category of environmental stochasticity. A small fraction of the variance is associated with the sampling process and some of it could be due to measurement errors. The main sources of environmental stochasticity have not been quantified but could include heavy rain and high winds at night, changes in wind direction and strength when mosquitoes are searching for a blood meal, breeding sites appearing and disappearing around the study sites, and pulses of mosquitoes emerging from water bodies. The quantity relevant for malaria transmission is the fraction of bites received by houses with the highest biting weights. Our study suggests the Pareto fraction for super-spreaders was on average closer to 66-33, ranging from 90-10 down to below 50-50. Though these are less heterogeneous than the proposed 80-20 rule, household heterogeneity would still have quantitative effects on transmission, invasion, exposure and immunity, the design of malaria studies, and the prospects for targeting. Our analysis suggests that the effects of super-spreading and super-spreaders would be strongest at low transmission intensity.

Translating general information about heterogeneity into a method for targeting would require finding covariates that would identify households with high biting weights and extrapolating these causes to establish relationships that could be used to target other households in the region. Some of this analysis has already been done for these sites; studies have found significant associations at some sites between the degree of urbanicity[33], elevation, household quality or ease of household entering[34], the number and age distribution of people living in a household, distance from water, and the enhanced vegetation index[30]. Collectively, these studies have established an evidence base and reasonable expectations about the gains in efficiency for targeting.

Since malaria immunity develops poorly and is only partially protective, and since most immune markers tend to be correlated with both exposure and immunity, immunity to malaria may be spuriously associated with disease[14–16]. These Pareto fractions, though not directly relevant for epidemiological studies of malaria or for disentangling cause and effect because they describe overall patterns without identifying who in a population is most exposed, can be used to inform and improve the design of epidemiological studies across the spectrum of transmission. Biting weights, which

do convey information about relative rates of exposure, can be estimated with some effort and when considered along with environmental and sampling noise, provide a more accurate measure of individual exposure that can be used to validate other metrics of exposure or to disentangle cause and effect in studies of the protective effects of malaria immunity. Biting weights are thus potentially useful for the proper design of epidemiological studies of malaria transmission, though our study suggests they have a weak overall effect.

These ideas could be applied to other infectious disease systems. Observed heterogeneity is likely multifactorial, and while we found important effects of seasonality and environmental stochasticity, other concerns may be important for pathogens with other modes of transmission. Heterogeneity is a basic feature of infectious disease transmission, but without some quantitative understanding of heterogeneity across the spectrum of transmission intensity, there is only a weak basis for understanding its consequences on pathogen transmission dynamics or control. Our analysis provides some basis for including heterogeneous biting in mechanistic models of mosquito-borne pathogens; for proper study designs and analysis of epidemiological data; and for evaluating the prospects of improving disease control through targeting. These other factors must be considered in understanding the consequences of heterogeneous biting for malaria and heterogeneous transmission, and more generally for infectious disease epidemiology, transmission dynamics, and control.

## Methods

**Data.** We examined heterogeneous transmission of malaria in a recent study at three sites with markedly different levels of malaria transmission in Uganda (Figs 1, 2); Anopheles gambiae s.l. and An. funestus s.l. are the dominant vectors, and a substantial fraction of exposure occurs at night in or around the home[29,34,35]. The study followed 330 houses monthly for 42 months in two locations and 69 months in one location. All children 0.5–10 years of age from each household were enrolled and followed longitudinally over the whole course of the study or until reaching the age of 11. All household members were given a long-lasting insecticidal net at the time of enrollment; reported bed net usage was high (98%), long-lasting, and consistent across study sites. Mosquitoes were caught in a Centers for Disease Control (CDC) light trap placed next to the bed in a room where at least one child participant was sleeping. The trap was placed indoors in the evening and retrieved the next morning, and mosquitoes were sorted, counted, and up to 50 per household were tested for sporozoites using an enzyme-linked immunosorbent assay (ELISA)[29]. Counts by CDC light traps were compared with other trapping methods and were highly correlated, albeit with different totals[29]. Ethical approval for this study was provided by the Uganda National Council for Science and Technology, the Makerere University School of Medicine Research and Ethics Committee, the University of California, San Francisco Committee on Human Research, London School of Hygiene and Tropical Medicine ethical committee and the School of Health and Biomedical Sciences Ethics Committee, Durham University[29,35].

The HBR and the EIR are entomological measures of malaria exposure: the number of bites by vector mosquitoes and by infectious vector mosquitoes, per person, per day. Here, we take the CDC light trap counts of anopheline mosquitoes and infectious anopheline mosquitoes as our estimated HBR and EIR. To compute the EIR, the HBR is multiplied by the sporozoite rate (SR). For households where more than 50 mosquitoes were counted in a night, EIR data were computed using the observed HBR multiplied by the sporozoite rate (proportion of infectious anopheline mosquitoes) from the sample of 50. For the period of October 2013 to March 2014, for which there were low overall mosquito counts and no sporozoite-positive mosquitoes, SR is estimated as the average of the 2 previous years: for example, October 2011 and 2012 were averaged to give an estimate for the October 2013 SR. This fraction was applied only to untested mosquitoes.

**Seasonality and biting weights.** To identify super-spreaders, we also conducted a statistical analysis of the data to quantify seasonality, biting, and the residual errors. Mosquito counts $Y = \{y_{h,d,n}\}$ for the $h$th household on sampling day $d$ at site $n$ are assumed to arise from a zero-inflated negative binomial distribution,

$$P\left(Y = y_{h,d,n}\right) = \begin{cases} \pi_{h,d,n} + \left(1 - \pi_{h,d,n}\right)\left(1 + \frac{\lambda_{h,d,n}}{\tau}\right)^{-\tau}, & y = 0, \\ \left(1 - \pi_{h,d,n}\right)\frac{\Gamma\left(y_{h,d,n} + \tau\right)}{y_{h,d,n}!\Gamma(\tau)}\left(1 + \frac{\lambda_{h,d,n}}{\tau}\right)^{-\tau}\left(1 + \frac{\tau}{\lambda_{h,d,n}}\right)^{-y_{h,d,n}}, & y > 0, \end{cases}$$

(1)

where $\tau > 0$ is a shape parameter quantifying the amount of overdispersion. Note that $P(Y = y_{h,d,n})$ is a function of $\pi_{h,d,n}$, $\lambda_{h,d,n}$, and $\tau$. The parameters $\pi_{h,d,n}$

and $\lambda_{h,d,n}$ can be modeled as a function of a set of explanatory variables. For $\pi_{h,d,n}$ it is common to use a logistic regression with a logit link function, as it describes a binomial process. A log link function is used to model the dependence of $\lambda_{h,d,n}$ on a different (or same) set of covariates. The log link function ensures that the estimated $\lambda_{h,d,n}$ will not be negative, regardless of parameter values. In our case, we model the dependence of $\pi_{h,d,n}$ and $\lambda_{h,d,n}$ on the same set of explanatory variables.

$$\text{logit}\left(\pi_{h,d,n}\right) = a_{h,n}\text{ID}_{h,n} + f\left(t_{d,n}\right) + \epsilon_{h,d,n} \quad (2)$$

$$\log\left(\lambda_{h,d,n}\right) = a_{h,n}\text{ID}_{h,n} + f\left(t_{d,n}\right) + \epsilon_{h,d,n}. \quad (3)$$

Here $\text{ID}_{h,n}$ denotes the $h$th household at site $n$; $a_{h,n}$ quantifies the effects of biting of the $h$th household at site $n$; $t_{d,n}$ denotes sampling day $d$ at site $n$ which is modeled as random effects; and the random effects $\epsilon_{h,d,n}$ are environmental stochasticity and sampling errors (a random variable that is Gaussian independently and identically distributed). The additional variation among mosquito counts that is not accounted for by the covariates is modeled via $\epsilon_{h,d,n}$, which in this model was Gamma distributed. The available time points (sampling days, $t_{d,n}$) are modeled as temporally structured random effects, ensuring that contiguous periods are likely to be similar, but allowing for flexible shapes in the evolution curve. First- and second-order random walks, and autoregressive processes of order 1 and order 2 are considered for modeling $t_{d,n}$[36]. The regression models are implemented in an integrated nested Laplace approximation framework via the R-INLA package[37]. For convenience and consistency, default prior specifications in R-INLA have been chosen for each of the prior distributions. A related simulation study was conducted to identify methods that could accurately estimate heterogeneity. This method—analysis using a zero-inflated negative binomial model[38]—was shown to give accurate estimates of biting weights, seasonal patterns, and environmental stochasticity[30]. The output of this analysis was a set of estimated quantities describing a biting weight for each house at each site, $\omega_{h,n} = \exp(a_{h,n})$, and a daily, site-specific expected number of bites by mosquitoes (i.e., the HBR), $S_{d,n} = \exp(t_{d,n})$. In Fig. 2a, the reported values of $\omega$ are shown along with the results of an MLE analysis for a Gamma distribution constrained to have a mean of 1, Gamma($\theta, \theta$), using the "shape" and "rate" parameterization in R.

**Variance components analysis.** To understand the distribution, we reformulated the negative binomial model (described above) as a Gamma-Poisson mixture process, where $y \sim \text{Poisson}(S_{d,n}\omega_{h,n}\varepsilon)$ and where $\varepsilon$ was Gamma distributed with mean of one. For each site, we estimated the proportion of the variance explained by each factor by computing the variance, and then sequentially the mean sum of squared differences between (1) the observations $y_{h,d,n}$ and $S_{d,n}$; and (2) the observations $y_{h,d,n}$ and $\omega_{h,n}S_{d,n}$. To estimate the fraction of remaining variance that was due to the sampling variance, we simulated counts from a Poisson process with mean $\omega_{h,n}S_{d,n}$, and compared the mean sum of squared differences for this simulated data and for the actual observations. This was repeated 1000 times to get a mean ratio. The average estimated ratio of these two quantities (and the 2.5th and 97.5th quantiles) was 0.133 (0.116–0.149) for Jinja, 0.032 (0.028–0.036) for Kanungu, and 0.017 (0.016–0.019) for Tororo, suggesting the sampling variance was a small part of the total remaining variance for Kanungu and Tororo.

**Pareto analysis.** As a first measure of heterogeneous biting, we use the Pareto fraction for super-spreading. The Pareto fraction for super-spreading for a given month is defined as the proportion of the total HBR or EIR and for which it is possible to say that a proportion $X$ with the highest HBR or EIR in that month got a proportion $1−X$ of the bites. The Pareto fraction for super-spreaders is defined as the proportion $X$ of the total HBR or EIR received by the proportion $1−X$ of households with the highest biting weights ($\omega_h$, see model description). This fraction is an intuitive measure of dispersion and it has been used as a simple way of describing the potential gains in efficiency if malaria control was targeted. The key difference between the Pareto fractions for super-spreading and for super-spreaders is that the latter measures biting by households that were consistently bitten most across the whole study, not just for the month.

We computed the Pareto fraction and index for each distribution from its empirical CDF (eCDF). Let $x$ index the observations sorted by their values; if $i < j$ then $x_i \geq x_j$. We let $X$ denote the proportion of $N$ observations and $Y$ denote the empirical cumulative distribution:

$$Y(X) = \sum_{i < XN} x_i. \quad (4)$$

For any distribution, $Y(X)$ is a monotonic increasing function that intersects the line $1−X$. The point $X$ where $Y(X) = 1−X$ defines the Pareto fraction for a distribution. At that point, there is a $X{:}1−X$ rule; for example, if $X = 0.9$, then there is a 90-10 rule. The Pareto index is

$$\log(X) / \log\left(\frac{X}{1-X}\right). \quad (5)$$

The modified Pareto fraction and index were computed in a similar fashion, but instead of sorting the observations directly, the observations are ordered by the associated biting weights, $\omega$; if $i < j$ then $\omega_i \geq \omega_j$. We let $W$ denote the fraction of $N$

observations computed in this way, and we compute the modified CDF,

$$Y(W) = \sum_{i < WN} \omega_i.$$

For any distribution, $Y(W)$ is a monotonic increasing function that intersects the line $1-W$. The point $W$ where $Y(W) = 1-W$ defines the modified Pareto value for a distribution. The modified Pareto index is

$$\log(W)/\log\left(\frac{W}{1-W}\right).$$

Notably, values of the modified Pareto distribution can violate the rules that generally apply to Pareto distributions. It is possible for the fraction of houses that tend to get the most bites (on average) to account for <50% of all bites in any particular month. A function in $R$ has been written that computes a Pareto summary for any set of observations, $y$, and their paired weights $\omega$; it has been included in the supplementary online data. The files to generate the figures have been archived along with this paper.

**Reporting summary.** Further information on research design is available in the Nature Research Reporting Summary linked to this article.

## Data availability

The primary data generated and analyzed during the current study are available at figshare (https://doi.org/10.6084/m9.figshare.6797408.v3). A table for each site containing a row for each observation: the count, the computed expected number of bites for that day, the biting weight for the individual is available for download (https://figshare.com/articles/Uganda_environmental_covariates_csv/6797408/3). Other data from the same study can be found in the Clinical Epidemiology Database Resources repository (https://clinepidb.org/ce/app). Primary input and output of this analysis is also included in a downloadable file called Figures.tgz from https://figshare.com/s/b273374317531a2e4377. All these files are found in the subdirectory Figures/inputs.

## Code availability

All code used for these analyses was also used for a previous paper. It is publicly available online at https://gatesopenresearch.s3.amazonaws.com/supplementary/12838/f7b1ea69-1fb5-46e5-98d8-5601c4d6f731.R. The analysis for this paper, including the code to do the Pareto analysis and to generate the figures is available for download in a single file, called Figures.tgz from https://figshare.com/s/b273374317531a2e4377. The figures can be remade in R by executing the file Figures/MASTER-SCRIPT.R. Scripts to create each one of the individual figures is found in the subdirectory Figures/scripts.

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

## Acknowledgements

We would like to thank those in the study communities who participated in this study or helped collect mosquitoes. Funding for the research and support for K.M., S.W.L., G.D., M.R.K., C.D., and D.L.S. came from National Institutes of Health as part of the International Centers of Excellence in Malaria Research (ICMER) program (U19AI089674). The authors wish to acknowledge the Infectious Diseases Research Collaboration (IDRC) for administrative and technical support. B.T.G., D.L.S., P.W.G., S.Y.K., S.W.L., D.B., and S.I.H. receive support from the Bill and Melinda Gates Foundation (105338, OPP1068048, OPP110495, OPP1106023, OPP1091919). P.W.G. is a Career Development Fellow (K00669X) jointly funded by the UK Medical Research Council (MRC) and the UK Department for International Development (DFID) under the MRC/DFID Concordat

agreement, also part of the EDCTP2 program supported by the European Union. S.I.H. is funded by a Senior Research Fellowship from the Wellcome Trust (095066).

## Author contributions

B.G., C.D., D.L.S., E.A., G.D., K.M., M.R.K., S.G.S., and S.W.L. designed and conducted the linked epidemiological and entomological PRISM studies. D.B., B.T.G., D.L.S., L.V.C., and S.Y.K conceived and designed the analysis. D.L.S., L.V.C., and SYK participated in the management and analysis of the data. D.L.S., L.V.C., and S.Y.K. wrote the first draft of the paper. I.R.B., B.G., P.W.G., R.C.R., P.E., and S.I.H. contributed to revisions of the paper in intermediate drafts. All authors participated in the writing of the final paper. All authors read and approved the final paper.

## Additional information

**Competing interests:** The authors declare no competing interests.

