## [Peer Review File · Nature Communications]

Reviewers' comments:

Reviewer #1 (Remarks to the Author):

The authors have carried out a thorough analysis of data on exposure to malaria parasites at three sites in Uganda in order to evaluate the extent of inter-household heterogeneity and its relationship with transmission intensity.

There is a considerable literature documenting heterogeneity in patterns of malaria transmission, so the novelty of the manuscript is in the analytical approach, and in the claims about Pareto rules. The conclusions are plausible, and in particular, the findings that the 80:20 rule is an oversimplification and the suggestion that transmission is more concentrated in a few 'super-spreaders' at low transmission make sense.

As the authors point out on page 9 (lines 133 et seq), the variation in exposure measured at household level (V) (reviewer's notation) can, in principle, be partitioned into effects of seasonality (V_s), household variation (V_h), environmental stochasticity (V_e) and sampling noise/measurement error (V_m) i.e. $V = V_s + V_h + V_e + V_m$. Figure 3b shows disaggregation of the total variance into three components (V_s , V_h , and $V_e + V_m$) by site, with the environmental and sampling noise grouped into one component. Is it possible to separate V_e and V_m , for instance by analyzing variation around outputs from the simulations referred to on line 327? What we are really interested in is $V_h/(V - V_m)$, i.e. the measurement error should not be part of the total that is being analyzed.

More critically, is it possible to carry out an analogous within-site analysis that goes beyond Figures 2b and 2d in allowing for measurement error? A between-site analysis to establish the trend with exposure needs to include far more than 3 sites, and the patterns in Figure 3b could easily be accounted for by random variation between sites, so it is the patterns of within-site variation by exposure that provide the key evidence for a decrease in exposure heterogeneity with increasing exposure, because the comparison here is across many different exposure levels.

The within-site analysis is presented in Figures 2b and 2d. The basis for these figures is implicitly model-based estimates of V_h/V . At low transmission, it is to be expected that the data are more sparse than at high transmission, so the proportion of the variance explained by sampling variation (V_m) might also be expected to be higher. It follows that the sampling variation should be subtracted out in the estimation of the proportion of variation attributable to each of the other components, in order to obtain a correct estimate of the trend with exposure. What we are interested in is $V_h/(V - V_m)$.

- If indeed $V_h/(V - V_m)$ decreases with increasing exposure, and V_m also decreases with increasing exposure, then V_h/V also decreases. So the qualitatively the conclusion from Figure 2 is likely to be robust, providing there is no bias in the estimation of V_h . However, if there is more measurement error at low exposure, then the trend with exposure would be steeper than that currently estimated.

- Conversely, if the model is assuming that measurement error is independent of exposure, then there could well be a bias in the opposite direction, since measurement error may be mis-assigned at low exposure as household variation, leading to an overestimate of V_h . It would be important to see convincing evidence that model assumptions about exposure dependence in measurement error cannot account for the trends in Figure 2.

Additional comments

P17. The paper uses the term 'Pareto Index' in a non-standard way; elsewhere this term is used for the exponent of the power-law underlying a Pareto distribution, which should be > 1 . For example, the 80–20 (4:1) rule corresponds to a Pareto Index of $\log(5)/\log(4) \approx 1.16$. In this sense, the quantity plotted on the vertical axis of Fig 2b is not the Pareto Index.

It is not very clear what is added by the analysis of sporozoite rates, or by the separate presentation of results for the HBR and EIR in Figure 2. Within the R-INLA framework described on p17 it should be possible to analyze the disaggregated sporozoite data using a Bayesian hierarchical model that correctly allows for the uncertainty induced by pooling of mosquitoes, and provides estimates for periods and locations with no sporozoite positive mosquitoes. 'Averaging over the sampling stochasticity' (line 280) across sites and periods would seem to defeat the objective of analyzing all the components of variation (one of which is the sampling variation). When the sporozoite data have been aggregated, thus destroying the original variance structure, it is hard to see how the analysis of the components of variation of the EIR can add any insights to the analysis of HBR.

It is rather unclear whether the quantity S is mosquito density or EIR. From the caption to Figure 1 it looks like it should be mosquito density, but this is not clear from the methods (line 331). If this is mosquito density, what is the notation for the EIR?

Line 46. The paper cited as reference(1) certainly discusses the 'Pareto Rule' but I cannot find any reference to it in reference(2) (Anderson & May's book), although there is substantial discussion of transmission heterogeneity there.

The discussion of how estimates of Pareto ratios might be used by programs or in trial design (p14) is not very convincing. Even the very thorough analysis of mosquito densities used in this paper does not provide clear quantitative results that could directly feed into program design or trials. The most obvious ways of analyzing heterogeneity to identify locations or areas to target and/or to inform trial designs use frequencies of human infection or disease. Such data generally don't lend themselves to analysis using Pareto rules because the measurements are often in the form of binary responses, though maybe there is potential to develop Pareto rules for incidence rates or search radii.

Reviewer #2 (Remarks to the Author):

This article used data from epidemiological studies in Uganda to investigate the phenomenon of super-spreading in malaria. There are some great ideas in here, and in particular I like the analysis of how the degree of heterogeneity varies with mean numbers of mosquitoes. However, there are a few fairly major areas where the article would need a lot more attention.

Major comments

Existing literature

The authors have neglected to cite some of their own important papers published recently in Nature Communications and eLife.

1. Guelbeogo et al. Variation in natural exposure to anopheles mosquitoes and its effects on malaria transmission. eLife 2018

Using mosquito data from Burkina Faso, this paper calculated the Pareto index on an individual level.

2. Goncalves et al. Examining the human infectious reservoir for Plasmodium falciparum malaria in areas of differing transmission intensity. Nature Comms 2017

Using data from Burkina Faso and Kenya, this paper quantified how variation in mosquito biting

rates and gametocyte densities contribute to heterogeneity in transmission.

3. Rodriguez-Barraquer et al. Quantification of anti-parasite and anti-disease immunity to malaria as a function of age and exposure. eLife 2018

This paper used data from the same study in Uganda being analysed in the authors' article. Importantly, this is an analysis of the individual-level data on malaria in humans.

Data: mosquitoes

The authors have excellent longitudinal data on mosquitoes caught in households, which facilitates investigation of super-spreading at this level. However, super-spreading is usually analysed as an individual-level phenomenon. Individual-level effects can be investigated through genetic matching of blood meals to household members, something done recently in another study site by a team led by one of this paper's authors (Goncalves et al. & Guelbeogo et al.). It's okay not to have this additional data on the individual-level, the authors just need to be explicit that their findings hold for super-spreading households.

Data: humans

In malaria, the phenomenon of super-spreading is dependent on both humans and mosquitoes. Super-spreaders will presumably be individuals with very high gametocyte densities (as shown nicely by Goncalves et al.). However, this analysis does consider any data from humans. This is a pity, since this data was collected and I believe some of it has already been published (Rodriguez-Barraquer et al.).

Data: sporozoite rate

"For the period of October 2013 to March 2014, for which there were low overall mosquito counts and no sporozoite-positive mosquitoes, SR is estimated as the average of the two previous years: for example, October 2011 and 2012 were averaged to give an estimate for the October 2013 SR."

This is not an appropriate thing to do. If the data say there were no sporozoite-positive mosquitoes, then you can't just replace it with data from another time period when there were sporozoite positive mosquitoes.

Figure 1

This plot shows an extension of the data presented by Kanya et al [ref 34]. However, comparing this to Kanya et al.'s Figure 5, I can't quite match the data up. Hard to say what's going on – perhaps some temporal smoothing effects?

Minor comments

Figure 2

I really like this figure, especially parts (b) and (d). The Pareto data (small circles) are visually dominated by the modified Pareto data (crosses).

Figure 3

I also like this figure, especially part (b). Some of the symbols appear to not have rendered correctly.

Tables and parameter estimates

There were no tables providing an overview of the data to be analysed. Also, there were no tables providing any values of the estimated parameters – not found in the text either.

Equations

The equations on lines 310 – 312 are really hard to follow. The function f has not been described. Later in the text there's mention of temporally structured random effects, but it's hard to know what's been done.

Inference

INLA was used for statistical inference. This can be considered a fairly heavy duty method. The authors should indicate why this method was selected, and since it is an approximate method they should provide some indication of how good the approximation is – I've not used INLA myself, but believe it can provide excellent approximations.

Kang et al. [ref 37] paper

A lot of the methods and data appear related to those also described in a recently published paper by Kang et al. which is only briefly mentioned at the very end of the paper.

Author contributions

For a paper of this length, there are a lot of authors (18), many of whom had no specific mention in the 'Author contributions'.

Response to Reviewers

Overall

Reviewer #1 (Remarks to the Author):

Comment 1.1

The authors have carried out a thorough analysis of data on exposure to malaria parasites at three sites in Uganda in order to evaluate the extent of inter-household heterogeneity and its relationship with transmission intensity.

There is a considerable literature documenting heterogeneity in patterns of malaria transmission, so the novelty of the manuscript is in the analytical approach, and in the claims about Pareto rules. The conclusions are plausible, and in particular, the findings that the 80:20 rule is an oversimplification and the suggestion that transmission is more concentrated in a few 'super-spreaders' at low transmission make sense.

As the authors point out on page 9 (lines 133 et seq), the variation in exposure measured at household level (V) (reviewer's notation) can, in principle, be partitioned into effects of seasonality (V_s), household variation (V_h), environmental stochasticity (V_e) and sampling noise/measurement error (V_m) i.e. $V = V_s + V_h + V_e + V_m$. Figure 3b shows disaggregation of the total variance into three components (V_s , V_h , and $V_e + V_m$) by site, with the environmental and sampling noise grouped into one component. Is it possible to separate V_e and V_m , for instance by analyzing variation around outputs from the simulations referred to on line 327? What we are really interested in is $V_h / (V - V_m)$, i.e. the measurement error should not be part of the total that is being analyzed.

More critically, is it possible to carry out an analogous within-site analysis that goes beyond Figures 2b and 2d in allowing for measurement error? A between-site analysis to establish the trend with exposure needs to include far more than 3 sites, and the patterns in Figure 3b could easily be accounted for by random variation between sites, so it is the patterns of within-site variation by exposure that provide the key evidence for a decrease in exposure heterogeneity with increasing exposure, because the comparison here is across many different exposure levels.

The within-site analysis is presented in Figures 2b and 2d. The basis for these figures is implicitly model-based estimates of V_h / V . At low transmission, it is to be expected that the data are more sparse than at high transmission, so the proportion of the variance explained by sampling variation (V_m) might also be expected to be higher. It follows that the sampling variation should be subtracted out in the estimation of the proportion of variation attributable to each of the other components, in order to obtain a correct estimate of the trend with exposure. What we are interested in is $V_h / (V - V_m)$.

- If indeed $V_h/(V - V_m)$ decreases with increasing exposure, and V_m also decreases with increasing exposure, then V_h/V also decreases. So the qualitatively the conclusion from Figure 2 is likely to be robust, providing there is no bias in the estimation of V_h . However, if there is more measurement error at low exposure, then the trend with exposure would be steeper than that currently estimated.
- Conversely, if the model is assuming that measurement error is independent of exposure, then there could well be a bias in the opposite direction, since measurement error may be mis-assigned at low exposure as household variation, leading to an overestimate of V_h . It would be important to see convincing evidence that model assumptions about exposure dependence in measurement error cannot account for the trends in Figure 2.

Response to 1.1

To estimate the sampling variance, we drew samples from a Poisson with mean: $P_{d,h} = \text{rpois}(S_d * w_h)$, and we compared $\sum (P_{d,h} - y_{d,h})^2$ to $\sum (S_d w_h - y_{d,h})^2$ to get an approximate ratio of the sampling variance, where S_d is the seasonal signal, w_h is the household weight, and $y_{d,h}$ is the observed HBR. This is now described in the Methods.

Comment 1.2

P17. The paper uses the term 'Pareto Index' in a non-standard way; elsewhere this term is used for the exponent of the power-law underlying a Pareto distribution, which should be > 1 . For example, the 80–20 (4:1) rule corresponds to a Pareto Index of $\log(5)/\log(4) \approx 1.16$. In this sense, the quantity plotted on the vertical axis of Fig 2b is not the Pareto Index.

Response to 1.2

In our original manuscript, we used “Pareto index” to mean the fraction of the total that could be accounted for by the top quintile. As reviewer #1 pointed out, we were using the term in a non-standard way. Under a change, the “modified Pareto index,” would become an even more confusing term. We have, therefore, changed our terminology to use Pareto index in a standard way, resulting in the following changes:

- The Pareto index is used in the standard way, and we now plot it. Figure 4 (old Fig2) has two extra panels showing the Pareto Index, computed in the standard way.
- Pareto fraction is the fraction $P:1-P$ (e.g. 80-20 or 90-10).
- We have changed our terminology to describe a “Pareto fraction for superspreading” and a “Pareto fraction for superspreaders.”

This required editing several parts of the document.

Comment 1.3

It is not very clear what is added by the analysis of sporozoite rates, or by the separate presentation of results for the HBR and EIR in Figure 2. Within the R-INLA framework described on p17 it should be possible to analyze the disaggregated sporozoite data using a Bayesian hierarchical model that correctly allows for the uncertainty induced by pooling of mosquitoes, and provides estimates for periods and locations with no sporozoite positive mosquitoes. 'Averaging over the sampling stochasticity' (line 280) across sites and periods would seem to defeat the objective of analyzing all the components of variation (one of which is the sampling variation). When the sporozoite data have been aggregated, thus destroying the original

variance structure, it is hard to see how the analysis of the components of variation of the EIR can add any insights to the analysis of HBR.

Response to 1.3

In this revised analysis, we have used the raw EIR counts data. The modeled EIR counts now apply only to the counts data from 699 (17%) samples in Tororo, 47 (1.2%) samples in Kanungu, and 3 (<0.1%) samples in Jinja, and no averaged sporozoite rate is used. To be consistent (and to make counts comparable) we have still extrapolated to get a measure of counts for those samples with >50 mosquitoes.

Comment 1.4

It is rather unclear whether the quantity S is mosquito density or EIR. From the caption to Figure 1 it looks like it should be mosquito density, but this is not clear from the methods (line 331). If this is mosquito density, what is the notation for the EIR?

Response to 1.4

We have clarified that it is the HBR at several places in the text.

Comment 1.5

Line 46. The paper cited as reference(1) certainly discusses the 'Pareto Rule' but I cannot find any reference to it in reference(2) (Anderson & May's book), although there is substantial discussion of transmission heterogeneity there.

Response to 1.5

We have deleted the reference to (2) at this point in the text.

Comment 1.6

The discussion of how estimates of Pareto ratios might be used by programs or in trial design (p14) is not very convincing. Even the very thorough analysis of mosquito densities used in this paper does not provide clear quantitative results that could directly feed into program design or trials. The most obvious ways of analyzing heterogeneity to identify locations or areas to target and/or to inform trial designs use frequencies of human infection or disease. Such data generally don't lend themselves to analysis using Pareto rules because the measurements are often in the form of binary responses, though maybe there is potential to develop Pareto rules for incidence rates or search radii.

Response to 1.6

Our analysis suggests that it should be considered, particularly as a factor in trial design. Because the environmental stochasticity has such a big effect, targeting super-spreaders is not expected to have large effects. To be perfectly clear, we did find an effect, and it could be important for study design, but our comments were meant to be circumspect, perhaps dampening the over-enthusiasm found previously and by others.

Reviewer #2 (Remarks to the Author):

Comment 2.1

This article used data from epidemiological studies in Uganda to investigate the phenomenon of super-spreading in malaria. There are some great ideas in here, and in particular I like the analysis of how the degree of heterogeneity varies with mean numbers of mosquitoes. However, there are a few fairly major areas where the article would need a lot more attention.

Major comments

Existing literature

The authors have neglected to cite some of their own important papers published recently in Nature Communications and eLife.

1. *Guelbeogo et al. Variation in natural exposure to anopheles mosquitoes and its effects on malaria transmission. eLife 2018*

Using mosquito data from Burkina Faso, this paper calculated the Pareto index on an individual level.

2. *Goncalves et al. Examining the human infectious reservoir for Plasmodium falciparum malaria in areas of differing transmission intensity. Nature Comms 2017*

Using data from Burkina Faso and Kenya, this paper quantified how variation in mosquito biting rates and gametocyte densities contribute to heterogeneity in transmission.

3. *Rodriguez-Barraquer et al. Quantification of anti-parasite and anti-disease immunity to malaria as a function of age and exposure. eLife 2018*

This paper used data from the same study in Uganda being analysed in the authors' article. Importantly, this is an analysis of the individual-level data on malaria in humans.

Response to 2.1

We thank the reviewers for drawing our attention to this oversight. We have now cited all these papers at the appropriate points in the discussion.

Comment 2.2

Data: mosquitoes

The authors have excellent longitudinal data on mosquitoes caught in households, which facilitates investigation of super-spreading at this level. However, super-spreading is usually analysed as an individual-level phenomenon. Individual-level effects can be investigated through genetic matching of blood meals to household members, something done recently in another study site by a team led by one of this paper's authors (Goncalves et al. & Guelbeogo

et al.). It's okay not to have this additional data on the individual-level, the authors just need to be explicit that their findings hold for super-spreading households.

Response to 2.2

We have clarified throughout the manuscript that our findings apply to households, but may not hold at the individual level.

Comment 2.3

Data: humans

In malaria, the phenomenon of super-spreading is dependent on both humans and mosquitoes. Super-spreaders will presumably be individuals with very high gametocyte densities (as shown nicely by Goncalves et al.). However, this analysis does consider any data from humans. This is a pity, since this data was collected and I believe some of it has already been published (Rodriguez-Barraquer et al.).

Response to 2.3

We hope to clarify that the purpose of this study was to characterize one dimension of entomological exposure. There is some work to be done relating exposure and infection, but it would not be appropriate for this particular study. The analysis of human infections by Rodriguez-Barraquer (2018) focused only on children. It examines malaria incidence and development of anti-parasite immunity as well as fever tolerance, but no data were collected on adults. Quantitative measures of gametocyte carriage were not collected.

It is certainly true that super-spreading also involves differences in net infectiousness, but this was not a question we could address with our data. That being said, we wish to emphasize that gross differences in net infectiousness are generally associated with age and anti-parasite immunity. Most of the evidence suggests that gametocyte densities are extremely variable over the course of a single infection, and that infectiousness is an extremely noisy process even after controlling for gametocyte densities. Aside from the patterns caused by anti-parasite immunity associated with age, we are not aware of any longitudinal studies suggesting some individuals are consistently more infectious than others.

There are well-documented differences among individuals in biting rates because of age and body size, but there is also almost certainly a great deal more heterogeneity in exposure given the activity patterns of individuals in the study. Some of these people will spend more time outside in the evening, when mosquitoes are active. It is certainly possible to model those relationships, but that would go beyond analysis of the data that we have.

This paper has focused on the patterns of exposure to the bites of vector mosquitoes at the household level in the household, because these are the data we have. We emphasize that this is the most extensive data describing longitudinal patterns of entomological exposure on households that has ever been collected, and so they are best suited to look at the issue of gross patterns of exposure being addressed in this paper.

Comment 2.4

Data: sporozoite rate

“For the period of October 2013 to March 2014, for which there were low overall mosquito counts and no sporozoite-positive mosquitoes, SR is estimated as the average of the two previous years: for example, October 2011 and 2012 were averaged to give an estimate for the October 2013 SR.”

This is not an appropriate thing to do. If the data say there were no sporozoite-positive mosquitoes, then you can't just replace it with data from another time period when there were sporozoite positive mosquitoes.

Response to 2.4

We have modified the analysis to examine the EIR counts data.

Comment 2.5

Figure 1

This plot shows an extension of the data presented by Kanya et al [ref 34]. However, comparing this to Kanya et al.'s Figure 5, I can't quite match the data up. Hard to say what's going on – perhaps some temporal smoothing effects?

Response to 2.5

These are, indeed, the same data but they have been plotted differently.

- The analysis by Kanya et al. is for one year of data, and it's showing the counts for positive mosquitoes.
- The plots in Figure 1 are the modeled expectations for HBR (daily expectation, across the population)

Comment 2.6

Figure 2

I really like this figure, especially parts (b) and (d). The Pareto data (small circles) are visually dominated by the modified Pareto data (crosses).

Response to 2.6

Thanks! We have modified this Figure so that the Pareto data are no longer visually dominated.

Comment 2.7

Figure 3

I also like this figure, especially part (b). Some of the symbols appear to not have rendered correctly.

Tables and parameter estimates

There were no tables providing an overview of the data to be analysed. Also, there were no tables providing any values of the estimated parameters – not found in the text either.

Response to 2.7

All of these results are from the analysis of the counts data, explained at the end of the methods section. These data are available for download from the website. The “estimated parameters” are an MLE fit to the household biting weights. This is now clearly explained in the text.

Comment 2.8

The equations on lines 310 – 312 are really hard to follow. The function f has not been described. Later in the text there’s mention of temporally structured random effects, but it’s hard to know what’s been done.

Response to 2.8

The “ f ” here is standard usage to denote a random effect associated with the day.

Comment 2.9

Inference

INLA was used for statistical inference. This can be considered a fairly heavy duty method. The authors should indicate why this method was selected, and since it is an approximate method they should provide some indication of how good the approximation is – I’ve not used INLA myself, but believe it can provide excellent approximations.

Kang et al. [ref 37] paper

A lot of the methods and data appear related to those also described in a recently published paper by Kang et al. which is only briefly mentioned at the very end of the paper.

Response to 2.9

We agree that INLA is heavy duty. INLA was used because we needed a way of guaranteeing that our household biting weights were accurate. A simulation study using INLA was described in the Gates Open Research paper.

Comment 2.10

Author contributions

For a paper of this length, there are a lot of authors (18), many of whom had no specific mention in the ‘Author contributions’.

Response to 2.10

All the authors listed on this manuscript have contributed to the paper, either through important discussions in formulating the paper, drafting the paper, or through their comments and modifications of earlier drafts

REVIEWERS' COMMENTS

Reviewer #1 (Remarks to the Author):

I am not 100% convinced that the response to point 1.1 of my initial review fully addresses the issue of identifiability of the measurement error. The authors' response refers to the section starting on line 385 that addresses this (which incidentally, I think should be headed 'Variance Components Analysis', not 'Analysis of Variance'). It would be helpful to see some text explaining the distributional assumptions and equations for the different quantities here. It seems that there is an assumption that the counts of mosquitoes biting any given human on a particular night are Poisson distributed, which is very questionable.

Reviewer #1 further comments

It is not so obvious whether differential attractiveness of hosts to mosquitoes (e.g. because of differences in body odour) would count under 'super-spreading' (I would say yes), or merely as a factor contributing to transmission heterogeneity. Of course, multiple factors affect exposure to mosquitoes (e.g. house construction, time spent outside, use of bed nets), most of which can lead to variation at the individual level, but also to geographical clustering.

I don't think it is realistic to revise the models to account for extra-Poisson variation in mosquito bites, because this would require the authors to use a different family of distributions (e.g. the Negative Binomial), but the over-dispersion parameter would probably not be identifiable. It might be possible to address this with sensitivity analyses (i.e. assuming different degrees of overdispersion and seeing how much difference this makes to the results) but this might be computationally challenging. This would be a significant complication that would make an already challenging manuscript inaccessible to many readers, so the details of this would need to be reported only in supplementary information.

In summary:

- I think the distinction between effects of super-spreading and other drivers of transmission heterogeneity would need to be carefully defined/specified.
- The authors should be able to handle this with textual revisions, which should make explicit both the terminology and also what are the biological phenomena that they are accounting for, and which they are not able to resolve.
- Sensitivity analyses looking at implications of extra-Poisson variation in mosquito biting would be desirable, but I don't think it would be reasonable to insist on these.

Reviewer #2 (Remarks to the Author):

The epidemiological data and the statistical analyses are now fundamentally sound, following substantial improvements.

My major reservation is that the results have been massively oversold as providing evidence of superspreading. To put things in perspective, it is worth looking at the definition from the canonical paper by Jamie Lloyd-Smith published in Nature in 2005:

"We propose this general protocol for defining a superspreading event: (1) estimate the effective reproductive number, R , for the disease and population in question; (2) construct a Poisson distribution with mean R , representing the expected range of Z due to stochasticity without individual variation; (3) define an SSE as any infected individual who infects more than $Z(n)$ others, where $Z(n)$ is the n th percentile of the Poisson(R) distribution."

It is also important to note that this definition applies in cases where it is known who infects who, either via rigorous contact tracing (as in the SARS outbreak), or from sequencing.

Based on this definition, a common sense definition of a malaria superspreader would be a person (or mosquito) who infects substantially more people (or mosquitoes) than expected by chance alone.

In this analysis of malaria data:

- We don't know who infects who.
- It's unclear what the superspreaders are: people, mosquitoes, households?
- No data from humans has been presented, despite the fact that it was collected.

What is really being presented here is heterogeneity in household exposure to infectious mosquitoes, and not superspreading.

Reviewer #2 further comments:

They're not showing superspreading in any commonly accepted usage of the term.

I would encourage the authors to clarify their use of the Poisson distribution. The Poisson distribution would characterise mosquito counts in the absence of heterogeneity - as they've spent a lot of time thinking about heterogeneity I would assume they know this.

Point by Point Response

- 1. First, both of the referees have raised an important issue regarding the conclusions that are presented. They have advised that the data are not showing evidence of super-spreading of malaria infection, which is individual host level variation such as that arising from differences in the propensity to give rise to gametocytes, as you have reported in the manuscript. Instead, the data are looking at transmission heterogeneity, or specifically at the consequences of heterogeneity in densities of biting mosquitoes. In light of these comments, we feel that it is critical that the claims in the manuscript are revised to accurately reflect transmission heterogeneity and not evidence of super-spreading.*

We thank the reviewer for this comment. Our understanding of superspreading is that applies to any aspect of transmission that causes the distribution of offspring (i.e. the number of infected hosts per infected host) to be over-dispersed. This could be caused by any number of factors: greater rates of shedding, a carrier state or more generally a longer period of infectiousness, or from a propensity to mix. For example, greater sexual activity by some individuals leads to them causing a greater number of cases. Mosquito biting, like sexual mixing, is indeed a kind of “transmission heterogeneity,” and it is also part of superspreading. We note that this is consistent with the comments from both reviewers below. We added two new paragraphs and a new section, entitled “Super-spreaders and Superspreading” to clarify how we are using the terms in this manuscript:

Heterogeneity in transmission can be quantified and discussed in terms of super-spreaders and super-spreading. Super-spreading for malaria describes heterogeneous transmission in which some hosts would infect more hosts in the next generation than expected by chance alone. Super-spreading can happen if some hosts are more infectious than others; for malaria, this would mean having gametocyte densities that were higher, that were carried longer, that were accompanied by gametocyte-stage blocking immunity that was lower than the population averages. Alternatively, malaria super-spreading can happen because a host is bitten by more vectors. A common finding is that mosquito counts follow negative binomial distributions, which suggests super-spreading through heterogeneous biting is common for malaria.

Super-spreaders are hosts who would consistently be found super-spreading; they are either consistently more infectious (e.g. with high gametocytemia or low immunity), or they are consistently bitten more by mosquitoes. Since it takes two bites to transmit malaria parasites from a human back to other humans, populations with super-spreaders have built-in correlations that amplify transmission; super-spreaders are both more likely to become infected and more likely to infect others²⁵⁻²⁹. To identify and target super-spreaders, it is necessary to identify those individuals who are consistently bitten more than others.

Super-spreaders & Super-spreading

The mosquito counts in our study are described well by negative binomial distributions; the variance of the counts data was consistently much higher than the mean³⁰, consistent with super-spreading. To quantify super-spreaders vs super-spreading, we developed a statistical model for these negative binomial distributions (Methods), and we have summarized the patterns using Pareto fractions (Methods). The analysis to compute Pareto fractions was modified for super-spreaders. The Pareto fraction for super-spreading is defined as the proportion X of the total HBR or EIR received by the $1 - X$ proportion of households (i.e. the point where the empirical cumulative distribution function intersects the line $1-X$). For super-spreaders, the same algorithm is repeated but only after re-ordering the observations by the biting weights ($\omega_{h,n}$, Methods). In the modified Pareto analysis, it is possible that those who *tend* to get the most bites (on average) will sometimes account for less than 50% of all bites.

The point that we seem to have failed to communicate, especially with regards to the comments from Reviewer #2, is that the data are negatively binomially distributed (hence not Poisson), and thus there is superspreading. Our analysis does already take this into account.

2. *Second, both of the referees have advised that justification for the use of the Poisson distribution must be provided, in addition to the biological phenomena that you are accounting for and those that you are unable to resolve.*
3. *Finally, we do feel that it is important that additional sensitivity analyses, which look at implications of extra-Poisson variation in mosquito biting, are provided in a further revised version of the manuscript.*

We thank the reviewer for this comment. I fear this is a misunderstanding that it is entirely our fault for being telegraphic in the main body of the text.

In fact, our data do follow negative binomial distributions, and the statistical analysis does account for the extra-Poisson variation. In the statistical model, the Poisson distribution is actually part of a compound Gamma-Poisson mixture process, which gives rise to a negative binomial distribution. In detail:

- The expected number of mosquitoes that would be counted in the average house on any given day is S_d .
- Each household has a biting weight, ω_h that has an expected value of one across the population. The expected number of mosquitoes that would be counted in the h^{th} household is $S_d \omega_h$.
- In addition to these household biting weights, a random variable from a Gamma distribution is drawn, denoted ϵ . Once again, this distribution is constrained to have a mean of 1, so that it affects only the shape of the distribution. The expected number of mosquitoes for the h^{th} household on day d thus has a distribution $S_d \omega_h \epsilon$.

- Finally, the model draws a Poisson variate $X \sim \text{Poisson}(S_d \omega_h \epsilon)$. Since ϵ is gamma distributed, X follows a negative binomial distribution.

We revised the text in the methods to explicitly state this. We also mention now that the data do follow a negative binomial distribution on lines 432-433 in the tracked changes version of the text.

REVIEWERS' COMMENTS

Reviewer #1 (Remarks to the Author):

I am not 100% convinced that the response to point 1.1 of my initial review fully addresses the issue of identifiability of the measurement error.

We thank this reviewer for this comment. The last time we updated the manuscript, we racked our brains to find some way – any way – of separating the sampling noise from any sources of error. We don't believe the identifiability question can be resolved any further.

The authors' response refers to the section starting on line 385 that addresses this (which incidentally, I think should be headed 'Variance Components Analysis', not 'Analysis of Variance').

We thank this reviewer for this comment. We have updated the section title accordingly.

It would be helpful to see some text explaining the distributional assumptions and equations for the different quantities here. It seems that there is an assumption that the counts of mosquitoes biting any given human on a particular night are Poisson distributed, which is very questionable.

We thank the reviewer for this comment. We believe these were described in the methods, but we have added some text to the main text (section: Household biting Weights and Environmental Stochasticity) that we hope will clarify how the model works. We assumed that the mosquito counts on any given were night were Poisson, given the mean. In the model, the mean has a gamma distribution, so this is a Gamma-Poisson mixture, which would make it a negative binomial distribution.

Reviewer #1 further comments

It is not so obvious whether differential attractiveness of hosts to mosquitoes (e.g. because of differences in body odour) would count under 'super-spreading' (I would say yes),

We thank the reviewer for this comment and agree.

or merely as a factor contributing to transmission heterogeneity. Of course, multiple factors affect exposure to mosquitoes (e.g. house construction, time spent outside, use of bed nets), most of which can lead to variation at the individual level, but also to geographical clustering.

We think that any sensible definition of superspreading for mosquito-borne pathogens would take into account for all of these factors. Our new sections (see below) define what we mean by superspreading so that our readers can make their own judgments.

Heterogeneity in transmission can be quantified and discussed in terms of super-spreaders and super-spreading. Super-spreading for malaria describes heterogeneous transmission in which some hosts would infect more hosts in the next generation than expected by chance alone. Super-spreading can happen if some hosts are more infectious than others; for malaria, this would mean having gametocyte densities that were higher, that were carried longer, that were accompanied by gametocyte-stage blocking immunity that was lower than the population averages. Alternatively, malaria super-spreading can happen because a host is bitten by more vectors. A common finding is that mosquito counts follow negative binomial distributions, which suggests super-spreading through heterogeneous biting is common for malaria.

Super-spreaders are hosts who would consistently be found super-spreading; they are either consistently more infectious (*e.g.* with high gametocytemia or low immunity), or they are consistently bitten more by mosquitoes. Since it takes two bites to transmit malaria parasites from a human back to other humans, populations with super-spreaders have built-in correlations that amplify transmission; super-spreaders are both more likely to become infected and more likely to infect others²⁵⁻²⁹. To identify and target super-spreaders, it is necessary to identify those individuals who are consistently bitten more than others.

I don't think it is realistic to revise the models to account for extra-Poisson variation in mosquito bites, because this would require the authors to use a different family of distributions (*e.g.* the Negative Binomial), but the over-dispersion parameter would probably not be identifiable. It might be possible to address this with sensitivity analyses (*i.e.* assuming different degrees of overdispersion and seeing how much difference this makes to the results) but this might be computationally challenging. This would be a significant complication that would make an already challenging manuscript inaccessible to many readers, so the details of this would need to be reported only in supplementary information.

We thank the reviewer for this comment. As described above, our analysis already accounts for extra-Poisson variation. Once again, we apologize for being so telegraphic and have made this explicitly clear in the text.

In summary:

- I think the distinction between effects of super-spreading and other drivers of transmission heterogeneity would need to be carefully defined/specified.
- The authors should be able to handle this with textual revisions, which should make explicit both the terminology and also what are the biological phenomena that they are accounting for, and which they are not able to resolve.

We thank the reviewer for these comments and agree. We have clarified our definitions in the text.

- Sensitivity analyses looking at implications of extra-Poisson variation in mosquito biting would be desirable, but I don't think it would be reasonable to insist on these.

We thank the reviewer for this comment and their understanding. Our analysis currently includes sensitivity analyses.

Reviewer #2 (Remarks to the Author):

The epidemiological data and the statistical analyses are now fundamentally sound, following substantial improvements.

My major reservation is that the results have been massively oversold as providing evidence of superspreading. To put things in perspective, it is worth looking at the definition from the canonical paper by Jamie Lloyd-Smith published in Nature in 2005:

“We propose this general protocol for defining a superspreading event: (1) estimate the effective reproductive number, R , for the disease and population in question; (2) construct a Poisson distribution with mean R , representing the expected range of Z due to stochasticity without individual variation; (3) define an SSE as any infected individual who infects more than $Z(n)$ others, where $Z(n)$ is the n th percentile of the Poisson(R) distribution.”

We thank the reviewer for this comment. We have clarified what we mean by superspreading in the main text. In the case of mosquito-transmitted pathogens, we are able to directly observe biting – the epidemiologically relevant event. We had not explicitly stated that the data were negatively binomially distributed, but we have now done so.

We hope this better sets the stage for aggregated patterns described in this manuscript by the Pareto analysis.

The updated text reads:

Heterogeneity in transmission can be quantified and discussed in terms of super-spreaders and super-spreading. Super-spreading for malaria describes heterogeneous transmission in which some hosts would infect more hosts in the next generation than expected by chance alone. Super-spreading can happen if some hosts are more infectious than others; for malaria, this would mean having gametocyte densities that were higher, that were carried longer, that were accompanied by gametocyte-stage blocking immunity that was lower than the population averages. Alternatively, malaria super-spreading can happen because a host is bitten by more vectors. A common finding is that mosquito counts follow negative binomial distributions, which suggests super-spreading through heterogeneous biting is common for malaria.

Super-spreaders are hosts who would consistently be found super-spreading; they are either consistently more infectious (*e.g.* with high gametocytemia or low immunity), or they are consistently bitten more by mosquitoes. Since it takes two bites to transmit malaria parasites from a human back to other humans, populations with super-

spreaders have built-in correlations that amplify transmission; super-spreaders are both more likely to become infected and more likely to infect others²⁵⁻²⁹. To identify and target super-spreaders, it is necessary to identify those individuals who are consistently bitten more than others.

It is also important to note that this definition applies in cases where it is known who infects who, either via rigorous contact tracing (as in the SARS outbreak), or from sequencing. Based on this definition, a common sense definition of a malaria superspreader would be a person (or mosquito) who infects substantially more people (or mosquitoes) than expected by chance alone.

We thank the reviewer for this comment and clarification. We agree with this definition and was used as our working definition.

We hope the text makes it clear that there is superspreading (i.e. because the mosquito counts data are so highly aggregated), but that the question we are trying to answer is how much of it is due to superspreaders.

In this analysis of malaria data:

- We don't know who infects who.
- It's unclear what the superspreaders are: people, mosquitoes, households?
- No data from humans has been presented, despite the fact that it was collected.

What is really being presented here is heterogeneity in household exposure to infectious mosquitoes, and not superspreading.

We thank this reviewer for these comments. To clarify, our study did not collect data on gametocyte densities, gametocyte stage transmission blocking immunity, or transmission. Super spreaders, in this analysis, are people, but our data are collected at households. We are considering these household counts as the best possible surrogate for individual heterogeneity in biting. Heterogeneity in household exposure to mosquitoes is the best available direct measure of transmission. We respectfully disagree with the reviewer on this point.

Reviewer #2 further comments:

They're not showing superspreading in any commonly accepted usage of the term.

We thank the reviewer for this comment. As suggested by the reviewer, we have now clarified how we are using our terms.

The updated text reads:

Heterogeneity in transmission can be quantified and discussed in terms of super-spreaders and super-spreading. Super-spreading for malaria describes heterogeneous transmission in which some hosts would infect more hosts in the next generation than expected by chance alone. Super-spreading can happen if some hosts are more infectious than others; for malaria, this would mean having gametocyte densities that were higher, that were carried longer, that were accompanied by gametocyte-stage

blocking immunity that was lower than the population averages. Alternatively, malaria super-spreading can happen because a host is bitten by more vectors. A common finding is that mosquito counts follow negative binomial distributions, which suggests super-spreading through heterogeneous biting is common for malaria.

Super-spreaders are hosts who would consistently be found super-spreading; they are either consistently more infectious (*e.g.* with high gametocytemia or low immunity), or they are consistently bitten more by mosquitoes. Since it takes two bites to transmit malaria parasites from a human back to other humans, populations with super-spreaders have built-in correlations that amplify transmission; super-spreaders are both more likely to become infected and more likely to infect others²⁵⁻²⁹. To identify and target super-spreaders, it is necessary to identify those individuals who are consistently bitten more than others.

I would encourage the authors to clarify their use of the Poisson distribution. The Poisson distribution would characterise mosquito counts in the absence of heterogeneity - as they've spent a lot of time thinking about heterogeneity I would assume they know this.

We thank the reviewer for this comment. We have updated the text to clarify this point in the section titled Household Biting Weights and Environmental Stochasticity..

REVIEWERS' COMMENTS:

Reviewer #1 (Remarks to the Author):

The authors have responded appropriately to my earlier comments and I am happy with the current version of the manuscript.